# IL-17 Ligand and Receptor Family Members Are Differentially Expressed by Keratinocyte Subpopulations and Modulate Their Differentiation and Inflammatory Phenotype

**DOI:** 10.3390/ijms26072989

**Published:** 2025-03-25

**Authors:** Elisabetta Palazzo, Roberta Lotti, Marika Quadri, Carlo Pincelli, Alessandra Marconi

**Affiliations:** DermoLab, Department of Surgical, Medical, Dental and Morphological Sciences, University of Modena and Reggio Emilia, 41124 Modena, Italy; roberta.lotti@unimore.it (R.L.); marika.quadri@unimore.it (M.Q.); carlo.pincelli@unimore.it (C.P.); alessandra.marconi@unimore.it (A.M.)

**Keywords:** psoriasis, keratinocytes stem cells (KSC), early transit amplifying (ETA) cells, late transit amplifying cells (LTA), IL-17 variants, IL-17 receptors

## Abstract

Psoriasis is a chronic inflammatory skin disease characterized by dysregulation of the interleukin 17 (IL-17) signaling axis. Given that psoriasis development depends on keratinocyte stem cells and early progenitors’ sensitivity to differentiation, we analyzed IL-17 ligands and the expression and function of in a novel subset of keratinocyte subpopulations: keratinocyte stem cells (KSC) and early and late Transit Amplifying (ETA or LTA, respectively) cells. We found that all subpopulations expressed all IL-17 variants, predominantly in ETA and LTA. Conversely, IL-17 receptor expression resulted in more heterogeneity, with IL-17RA, -C, and -E being the most differentially regulated. Stimulus with IL-17A, IL-17-F, IL-17-A/F, and IL-17C promotes the upregulation of CXCL1, CXCL8, and DEFB4 mRNAs expression in both KSC and ETA. Moreover, IL-17A and IL-17A/F mainly decrease KSC proliferation and promote cell cycle block. Globally, IL-17A and IL-17A/F modulated the expression of proliferation, differentiation, and psoriasis-associated markers. Furthermore, KSC- and ETA-derived 3D reconstructions displayed increased epidermal thickness and upregulated KRT16 expression after treatment with IL-17A or IL-17A/F. Therefore, our data demonstrated that IL-17 family members perform distinctive functions in a specific keratinocyte subpopulation and define IL-17 signaling as a critical modulator of KSC behavior, proving its role in epidermal homeostasis dysregulation of psoriasis.

## 1. Introduction

Psoriasis is a highly prevalent immune inflammatory skin disease with both environmental and genetic components, the latter represented by ten or more distinct psoriasis susceptibility loci (PSORS) [1]. The pathogenesis of psoriasis is an object of intensive investigation, but many unanswered questions remain concerning the role of keratinocytes and immune cells. Whereas keratinocytes might be viewed only as bystanders in the context of deregulation of the immune system, it is more likely that they are active participants in the pathogenesis of psoriasis. Indeed, keratinocytes express major pathological changes associated with the disease. Moreover, they produce the highest amounts of antimicrobial peptides (AMPs), which are considered major players in innate immunity, which in turn cooperates with adaptive immunity in the mechanisms underlying psoriasis [2]. In addition, chemokines, such as interleukin 8 (IL-8) and chemokine (C-X-C motif) ligand 1 (CXCL1), produced by keratinocytes in the epidermis, act on both innate and acquired immune systems.

One of the most recent and promising developments in psoriasis research is the definition of the role of the Th17 CD4+ T cell and one of its cytokines, interleukin 17 (IL-17), in the disease pathophysiology, progression, and maintenance [3,4]. There are six well-defined IL-17 ligands and five receptors. The interleukin 17A (IL-17A), interleukin 17F (IL-17F), and interleukin 17C (IL-17C) ligands are highly expressed in psoriatic skin and their presumed major cellular sources in the skin are highlighted, although other sources may contribute. These three ligands and interleukin 17E (IL-17E), known as interleukin 25 (IL-25), have demonstrated heteromeric receptor complexes, which in all cases include the IL-17 receptor A subunit and one specific IL-17 receptor subunit partner. Major cytoplasmic factors interacting with the IL-17 receptor complex have been demonstrated for the Interleukin Receptor A (IL-17RA)/Interleukin Receptor C (IL-17RC) complex, where they are best studied [5]. The IL-17A and IL-17F ligands form homo- and heterodimeric complexes, while the ligand-receptor interactions are less well-defined for interleukin 17B (IL-17B) and interleukin 17D (IL-17D), and the requirement for an IL-17 receptor A subunit is unknown [5].

Finally, keratinocytes can carry defects, leading to psoriasis formation. Indeed, keratinocytes are characterized by altered differentiation, hyperproliferation, and a reduced rate of apoptosis [6]. Transgenic mice expressing α2β1 integrin have previously been demonstrated to spontaneously develop a skin disorder resembling psoriasis [7]. Teige and colleagues recently showed that mild epidermal wounding induces uniform acanthosis and an influx of immune cells in these animals. In integrin transgenic mice, chronic inflammation develops with substantial keratinocyte hyperproliferation, inflammatory infiltration, and high cytokine levels. This suggests that altered keratinocyte differentiation and proliferation can drive skin inflammation and cause chronic immune cell activation. Keratinocytes may be important in initiating, sustaining, and amplifying inflammatory responses by expressing molecules involved in T cell recruitment, retention, and activation. Furthermore, it is not known which keratinocyte subpopulation carries the “psoriatic defect”, thus being responsible for the aberrant epidermal phenotype of the disease. In this respect, it has been proposed that epidermal abnormalities in psoriasis are due to a defect in the transit amplifying (TA) compartment rather than a reduction in the cell cycle time [8]. This has recently been confirmed by in silico studies that simulate psoriasis by altering TA cells. Prolonging the proliferation time of TA cells leads to the main characteristics of psoriasiform morphology: an absolute increase in the number of proliferating cells, a marked increase in the differentiated compartment, a higher proportion of germinative cells, and a marked reduction in turnover time [9]. The inherent malfunction in the behavior of this subpopulation is enforced by altered levels of several markers in TA cells from psoriatic lesions compared to TA from normal skin [10]. Moreover, psoriasis is thought to be influenced by the differential sensitivity of keratinocyte stem cells (KSC) and TA to apoptosis, as Weatherhead et al. showed that UVB can cause a higher increase in apoptotic cells in TA than in KSC [11]. Lastly, in normal epidermis, the common neurotrophin receptor CD271, also known as p75NTR, is mostly confined to the TA cell subpopulation, while CD271 levels are significantly lower in psoriatic than in normal TA keratinocytes [12]. In addition, the rate of apoptosis in psoriatic TA cells is significantly lower than that in normal TA cells. CD271 ligands fail to induce apoptosis in psoriatic TA cells, and CD271 retroviral infection restores ligand-induced apoptosis in psoriatic keratinocytes [12]. We demonstrated that CD271 is critical for keratinocyte differentiation and regulates the transition from KSC to TA cells, identifying an early stage of TA keratinocytes [13]. More recently, we demonstrated that the expression of CD271 could identify a population of early progenitors, named Early Transit Amplifying (ETA) cells, with unique features, according to proliferation, differentiation, and colony-forming efficiency features [14].

Against this background, the study of keratinocyte subpopulations in relation to IL-17 effects is of fundamental importance for understanding psoriasis pathogenesis and progression.

## 2. Results

### 2.1. Evaluation of IL-17 Ligand and Receptor Family Members’ Expression in Keratinocyte Subpopulations

To evaluate the expression and function of IL-17 variants (IL-17A, IL-17B, IL-17C, IL-17D, IL-17E (IL25), and IL-17F) and their receptors in keratinocyte subpopulations, we first isolated KSC, ETA, and Late Transit Amplyfing (LTA) from healthy donors, as previously described [15]. By the analysis of the mRNA expression, we demonstrated that all three keratinocyte subpopulations express all the IL-17 variants (IL-17A, IL-17B, IL-17C, IL-17D, IL-17E (IL25), and IL-17F) (Figure 1a). In particular, IL-17 variants were predominantly expressed by more differentiated keratinocytes, with a statistically significant increase in both ETA and LTA. Moreover, LTA cells expressed the highest levels of all IL-17 variants compared to KSC and ETA. In contrast, the analysis of IL-17 receptor variant expression is more varied. Indeed, while IL-17RA, IL-17RC, and IL-17RE were significantly more expressed by transit amplifying keratinocytes (both ETA and LTA), with an increase in LTA cells, differences in IL-17RB and IL-17RD mRNA levels were not statistically significant in the three subpopulations (Figure 1b). Therefore, the expression of IL-17RA, IL-17RC, and IL-17RE could be potentially functional during keratinocyte differentiation, according to the higher expression of their ligands.

### 2.2. Evaluation of the Biological Response to IL-17A, IL-17F and IL-17A/F Treatment in Keratinocyte Sub-Populations

Given the previous results, to establish whether the expression of IL-17 receptors determines a differential response in keratinocyte subpopulations, we treated healthy human KSC, ETA, and LTA with human recombinant IL-17A, IL-17F, and IL-17A/F, and analyzed the modulation of the expression of known chemokine and antimicrobial peptide (AMP) mRNAs (Figure 2a).

As already shown in total keratinocytes [16,17], IL-17 treatment upregulates some chemokines (CXCL1 and chemokine (C-X-C motif) ligand 8 (CXCL8) or interleukin 8 (IL-8)) and some antimicrobial peptides (DEFB4). Here, we show that in KSC, IL-17A treatment induces an increase in both chemokines and AMP. Conversely, IL-17F predominantly upregulates *DEFB4* (RNA for human β-defensin-2), which is normally expressed at high levels in psoriatic lesions. IL-17 A/F synergistically upregulates *DEFB4* levels (Figure 2a). Analysis of ETA showed that IL-17 upregulated the expression of all the genes. In particular, IL-17A increases *CAMP* (RNA for LL37) in a very important way, while IL-17F acts predominantly on *CXCL8* and *DEFB4*. The IL-17A/F heterodimer synergistically modulates *CXCL8* and *CAMP* expression. Finally, while IL-17A had a minimal effect on LTA, IL-17F induced an increase in *DEFB4* expression. IL-17 A/F heterodimer synergistically increased CXCL1 expression only. IL-17 treatment dramatically upregulated *CXCL1* and *DEFB4* in KSC and *CXCL8* and *CAMP* in ETA cells, while LTA cells appeared less sensitive to IL-17 stimuli despite IL-17 receptor mRNA expression.

### 2.3. IL-17A, IL-17F, and IL-17A/F Treatment Differentially Modulates Viability and Cell Cycle in Keratinocyte Subpopulations

We have previously demonstrated that keratinocyte subpopulations express all IL-17 variants and their receptors, and treatment with human recombinant IL-17A, IL-17F, and IL-17A/F upregulates the expression of known chemokine and antimicrobial peptide (AMP) RNAs, such as *CXCL1*, *IL-8*, and *DEFB4*, mostly in KSC and ETA cells. To further determine the KSC, ETA, and LTA behavior after IL-17 treatment in terms of proliferation and differentiation, we performed a single treatment with human recombinant IL-17A, IL-17F, and IL-17A/F and analyzed their effects at different time points.

Cell proliferation was evaluated in KSC, ETA, and LTA cells after 24- and 72-h treatments (Figure 2b). As expected, the KSC and ETA subpopulations were mainly affected by IL-17 compared to LTA. In particular, KSC showed higher proliferation at 24 h after stimulation, followed by a statistically significant decrease in viability at 72 h. Similarly, ETA cells showed a lower proliferation trend at 72 h post-stimulation with all the analyzed IL-17 variants. Conversely, LTA cells were not affected by all IL-17 treatments at any time point. These results were also corroborated by the analysis of the cell cycle (Figure 2c), which globally indicated that IL-17 variants induce an increase in the G_0_G_1_ phase and a decrease in the S phase in KSC at 72 h. In contrast, the cell cycle phases of ETA and LTA were not significantly affected.

### 2.4. IL-17A and IL-17A/F Treatment Modulates Keratinocyte Subpopulation Proliferation and Differentiation Marker Expression

Given the major effect of IL-17A and IL-17A/F on keratinocyte subpopulation cell cycle and viability, we analyzed specific markers associated with epidermal differentiation and proliferation to further understand the IL-17A- and IL-17A/F-dependent effects on cell cycle and viability. First, we analyzed the expression of keratin 10 (KRT10) and involucrin as early and late differentiation markers, respectively. In normal epidermis, the first sign of differentiation occurs within the basal layer in a subpopulation of keratinocytes that start to express keratin 1 (KRT1) and KRT10 suprabasal keratin transcripts. However, in the psoriatic epidermis, there is a delay in the expression of these markers, which normally begin several suprabasal layers [18]. Moreover, we demonstrated that normal keratinocytes undergo aberrant expression of differentiation markers after ectopic expression of psoriasis-associated molecules such as E-FABP [19]. Involucrin, a precursor protein that plays a role in stabilizing the cornified envelope (CE), is also upregulated in psoriasis vulgaris [20], where CE formation appears to be initiated prematurely, and involucrin remains the major constituent of the CE during maturation. The analysis of KRT10 expression in KSC and ETA cells after stimulation with recombinant human IL-17A and IL-17A/F heterodimer revealed that all of them could induce and significantly upregulate the expression of such markers in keratinocyte subpopulations (Figure 3a). Similarly, involucrin expression appeared to be strongly induced in KSC by IL-17A and IL-17A/F heterodimers and upregulated in ETA and LTA cells by IL-17A and IL-17A/F heterodimers (Figure 3b).

Keratin 15 (KRT15) is a type I keratin protein co-expressed with the KRT5/KRT14 pair present in the basal epidermal layer, and it is associated with the keratinocyte proliferating compartment. However, the expression of KRT15 can be promoted by several conditions in vivo, such as the proinflammatory stimulus by phorbol ester PMA through protein kinase C (PKC)/activator protein 1 (AP1) signaling or by the activity of the transcription factor FOXM1 [21,22]. The analysis of KRT15 expression in keratinocyte subpopulations showed a substantial difference between KSC, ETA, and LTA in terms of their response to IL-17s (Figure 3c). In KSC, while IL-17A induces a downregulation in KRT15 expression, IL-17A/F significantly increases its expression. On the contrary, after treatment with all IL-17s, ETA cells showed no expression of KRT15, which is consistent with the previously observed increase in differentiation markers. Interestingly, LTA displayed significantly higher KRT15 expression after stimulation with all IL-17 variants. Similar to the response of keratinocytes to PMA [22], we speculated that an inflammatory stimulus induces the “activation status” in keratinocytes. However, given the complexity of these results, a deeper analysis of KRT15 modulation by IL-17s will be needed to clarify this mechanism.

Keratin 16 (KRT16) expression is induced in hyperkeratotic lesions as well as atopic dermatitis or psoriasis [23,24,25], and a specific role in the regulation of immune response has been proposed [26]. Our results (Figure 3d) indicated that KRT16 expression was significantly upregulated in KSC by IL-17A and IL-17A/F heterodimer, confirming the key role of such molecules in promoting the psoriatic phenotype and corroborating previous works indicating the upregulation of the stemness compartment features by IL-17 and interleukin 22 (IL-22) [27]. In contrast, KRT16 was increased in both ETA and LTA by IL-17A but decreased by IL-17A/F, indicating the different roles of such molecules in progressively more mature keratinocytes.

### 2.5. IL-17A and IL-17A/F Treatment Modulates the Expression of Psoriasis-Associated Markers in Keratinocyte Subpopulations

Our results revealed that IL-17 signaling and its increase in proinflammatory and psoriatic skin conditions significantly affect keratinocytes according to their involvement in the different epidermal compartments by modulating the expression of proliferation and differentiation markers, as well as cell viability in vitro. Therefore, we investigated the capacity of IL-17A and IL-17A/F, which were found to be the most effective molecules, to modulate or promote the expression of different markers that are notably upregulated in psoriatic skin, to discriminate a specific effect on KSC, ETA, or LTA cells.

Epidermal keratinocytes in psoriatic lesions are characterized by activated Signal Transducer and Activator of Transcription 3 (STAT3), and transgenic mice with a constitutive activation of Stat3 can develop a phenotype that resembles psoriasis [28]. In detail, nuclear localization of the activated phosphorylated form of STAT3 has been found in psoriatic plaques [28]. In the present study, we investigated STAT3-P levels in keratinocyte subpopulations, revealing that stimulation of KSC with IL-17-A and IL-17A/F can promote an increase in the number of cells expressing STAT3-P in their nuclei, thus supporting the promotion of proliferation in the stemness compartment (Figure 4a,b). A similar effect was observed with IL-17-A/F in LTA cells (Figure 4a,b). In contrast, ETA cells, which show a higher level of STA3-P, were not significantly affected by the treatments in this context. Given the role of early epidermal progenitors in the expansion of epidermal cells [14], they likely present unique activation pathways involving the STAT3-dependent axis.

Cutaneous defense activities are mediated not only by a physical barrier, but also through the production of antimicrobial molecules, including the AMPs (antimicrobial peptides), such as the S100 family proteins, and β-defensin [29]. S100A7, also known as psoriasin, is normally expressed in both healthy and psoriatic epidermis, with an increase in psoriatic skin and at the plasma membrane of the spinous layers [30,31,32]. Several studies have shown that S100A7 expression levels increase in response to inflammatory stimuli and are associated with epidermal accumulation of CD4+ lymphocytes and neutrophils [33,34]. In the context of keratinocyte subpopulations (Figure 4b), we confirmed the higher expression of LTA cells in the plasma membrane. IL-17A increased the number of positive KSC and ETA cells in both the cytoplasmic and plasma membranes, and a similar effect was observed with IL-17A/F in KSC. In contrast, the number of LTA cells expressing S100A7 was reduced by all IL-17 variants, but S100A7 positive cells showed major cytoplasmic localization (Figure 4b).

Therefore, as previously shown at the mRNA level (Figure 2), treatment with IL-17 variants can significantly upregulate the expression of AMPs. IL-17A and IL-17-A/F induce an increase in DEFB4 (RNA for human β-defensin-2), which is normally increased in psoriatic lesions [35], mostly on KSC. Here, we confirm these results at the protein level, showing an increase in β-defensin levels after treatment with IL-17 and IL-17-A/F in KSC (Figure 4c).

Globally, these results indicate that a psoriatic-like phenotype is promoted by IL-17 molecules acting on the stemness compartment, given the higher modulation of KSC features.

### 2.6. IL-17A and IL-17A/F Treatment Influences Subpopulation-Derived Skin Reconstruct Epidermal Thickness and Proliferation and Differentiation Marker Expression

It has been shown that a mixture of IL-17, IL-22, and TNF-α resulted in destabilization of the epidermis, parakeratosis, and hypogranulosis, resembling partially the lesional psoriasis phenotype [36], and several 3D models have been proposed to study psoriasis features [37]. Therefore, we further evaluated the effects of IL-17A and IL-17A/F heterodimers on the epidermal phenotype generated by KSC or ETA cells by creating a skin equivalent (Figure 5). By analyzing the histological features, we found that the addition of IL-17A is able to increase the epidermal thickness in both KSC- and ETA-derived skin reconstruction. The same effect was observed with IL-17A/F on ETA-derived epidermis. Moreover, the addition of interleukins promoted epidermal invagination, which was more pronounced in IL-17A-treated KSC and ETA skin and IL-17A/F-treated ETA skin. Therefore, owing to the addition of only a single stimulus, we were able to act on keratinocyte features and their cross-interaction in a 3D model, thus demonstrating the effect of such molecules primarily on keratinocytes in the absence of other cell types.

### 2.7. Evaluation of the Preliminary Biological Response to IL-17B, IL-17C, IL-17D and IL-17E Treatment in Keratinocyte Subpopulations

Our previous results indicated that the expression of IL-17 receptor variants in keratinocyte subpopulations varies (Figure 1b), and the biological response of KSC, ETA, and LTA to IL-17A, -A/F, or -F depends on the specific characteristics of the cell type. Furthermore, we found that the effects of IL-17-A and IL-17-A/F were comparable and mostly acted on the epidermal cell stem and proliferative compartments. IL-17-A, IL-17F, and IL-17-A/F are primarily involved in the development of psoriasis [4]. However, a differential role of IL-17B, IL-17C, IL-17D and IL-17E has been given during inflammation and, overall, in psoriasis [5,38,39]. IL-17B has been understudied in psoriasis, but it seems to be modulated by cytokines [40]. In contrast, IL-17D may be involved in interleukin 36 (IL-36)-mediated skin inflammation [41]. Given this background, we investigated the effects of IL-17B, IL-17C, IL-17D, and IL-17E on keratinocyte subpopulations. Our results indicated that they can effectively modulate the expression of *DEFB4*, *CXCL1*, *CXCL8*, *and CAMP* in keratinocyte subpopulations, but with different expression patterns (Figure 6). By analyzing mRNA expression, we found a complex scenario.

CXCL1 was significantly upregulated in KSC by IL-17C; we found a trend with a greater upregulation of this cytokine in KSC by all the analyzed IL-17s, most likely due to sample size limitations. CXCL8 was induced by IL-17C and IL-17E in KSC and by IL-17E in ETA. Similarly, DEFB4 was significantly upregulated in KSC and ETA by IL-17C treatment. Overall, we can highlight a strong effect of IL-17C on the induction of all the analyzed molecules, in agreement with a previous work showing the induction by IL-17C of β-defensin and G-CSF in stimulated keratinocytes [42]. As for CAMP, we found a significant upregulation by IL-17D and IL-17E in KSC and by IL-17C in ETA cells. Moreover, treatment with IL-17B, IL-17D, and IL-17E decreased CAMP levels in ETA cells.

## 3. Discussion

Psoriasis is a chronic immune-mediated inflammatory skin disease that is well characterized from both histological and clinical points of view. Our knowledge of its pathogenesis has increased enormously over the last 20 years and is still evolving. A number of relevant works have initially identified T cell accumulation in psoriatic lesions [43,44,45], and improvement of skin lesions and reduction of inflammatory infiltrates have been observed owing to a combination of novel and technologically advanced immunosuppressive drugs, such as the development of targeted monoclonal antibodies [46]. One of the most recent and promising developments in psoriasis research is the definition of the role of the Th17 CD4+ T cell and one of its cytokines, IL-17, in the disease’s pathophysiology, progression, and maintenance [3]. Nevertheless, the classical keratinocyte dysregulation hypothesis continues to contribute to a deep understanding of psoriasis pathogenesis at the molecular level.

In light of these considerations, the development of novel techniques for the isolation and studies of different epidermal compartments has led to the identification of a specific subset of cells within the epidermis with unique features. We recently obtained an enrichment of KSC, ETA, and LTA cells in vitro by taking advantage of their ability to adhere to collagen IV. Specifically, we identified ETA cells as early progenitors, in which the expression of CD271 determines the path to the maturation process [14]. In the absence of CD271, as reported in psoriatic skin and keratinocytes, a hyperproliferative phenotype with altered differentiation can develop [13].

In this context, we provide evidence for the expression and function of IL-17 variants according to the behavior of KSC, ETA, and LTA cells. Keratinocyte subpopulations not only express all IL-17 receptor variants, but these receptors also seem to be functional in the modulation of conventional chemokines (*CXCL1* and *CXCL8*) and AMPs (*DEFB4* and *CAMP*). While *IL-17RA*, *IL-17RC*, and *IL-17RE* were significantly more expressed by transit amplifying keratinocytes (both ETA and LTA), with an increase in LTA cells, differences in *IL-17RB* and *IL-17RD* RNA were not statistically significant in the three subpopulations. Therefore, our data are in agreement with the role of IL-17RA, IL-17RC, and IL-17RE in psoriasis, but we could highlight a differential expression pattern in the epidermis.

IL-17 acts predominantly on proliferating keratinocyte subpopulations [27]. Here, we identified a major role of IL-17 and IL-17A/F on KSC; however, the mechanism by which IL-17 modulates AMPs in the basal keratinocyte subpopulation remains to be determined, while these substances are mostly detected and increased in the upper epidermal layers.

Moreover, we found that IL-17B, IL-17C, IL-17D, and IL-17E can modulate the expression of *DEFB4*, *CXCL1*, *CXCL8,* and *CAMP* in keratinocyte subpopulations. Compared to IL-17A, IL-17F, and IL-17A/F heterodimers, IL-17B, IL 17C, IL-17D, and IL-17E are less characterized in the context of psoriasis. IL-17B has recently attracted researchers because of its role in tumors [47], while IL-17C can potentiate Th17 cell activity in regulating the proinflammatory response [48]. IL-17D is considered to be the most ancient cytokine in the IL-17D family [49] and can promote the immune response by recruiting NK cells. IL-17E (also known as IL-25) is a proinflammatory cytokine that favors Th2-type immune responses [50].

Our data reveal a major role of IL-17C in KSC, thus identifying, together with IL-17A and IL-17A/F, a subset of IL-17s whose action on the stemness compartment seems to be crucial for proliferation and differentiation. Therefore, our results suggest an important role for keratinocyte subpopulations, particularly KSC and ETA, in psoriasis modulation. Further studies are required to identify the molecular mechanisms that regulate IL-17-dependent KSC behavior during the pathogenesis of psoriasis.

Therefore, our data demonstrate that IL-17 family members perform distinctive roles in specific keratinocyte subpopulations. Globally, these results confirm and define IL-17 as a critical modulator of keratinocyte hyperproliferative subpopulations and prove its role in the dysregulation of epidermal homeostasis in psoriasis.

## 4. Materials and Methods

### 4.1. Human Keratinocyte Culture

Normal human epidermal keratinocytes (NHEK) were isolated from foreskin surgical samples of healthy skin. Tissues were obtained from Policlinic of Modena, Italy. All subjects gave their informed consent for inclusion before they participated in the study. The study was conducted in accordance with the Declaration of Helsinki, and the protocol was approved by the Ethics Committee of Modena, Italy (Prot. N. 184/10).

### 4.2. Keratinocyte Subpopulation Isolation and Skin Equivalents

Fresh keratinocytes were divided into KSC, ETA, and LTA cells based on their ability to adhere to type IV collagen (100 μg/mL; Sigma, St. Louis, MO, USA). Briefly, keratinocytes were first allowed to adhere to type IV collagen for 5 min (KSC), and the non-adherent cells were then transferred to fresh collagen-coated dishes and allowed to attach for 15 min (ETA cells), as previously reported [15]. In contrast, LTA cells were allowed to attach to a fresh collagen-coated plate overnight, and three keratinocyte populations were cultured in serum-free medium (KGM, Clonetics, San Diego, CA, USA).

Cells were treated with hr-IL-17A, hr-IL-17A/F, or hr-IL-17F (R&D Systems, Inc., Minneapolis, MN, USA) and after 24 or 72 h of treatment, cells were analyzed using RT-PCR, MTT assay, and immunofluorescence.

Skin equivalents were obtained by seeding KSC or ETA cells on dermal equivalents generated by fibroblast-induced type I collagen contraction, as previously described [15]. At each media change, specific rh-IL-17 was added in both submerged and air-exposed phases.

### 4.3. RT-PCR

RNA was isolated using the PureLink RNA Mini Kit (Ambion^®^, Thermo Fisher Scientific) according to the manufacturer’s instructions. On-column PureLink DNase Treatment was performed with PureLink DNase (Invitrogen, Thermo Fisher Scientific) to obtain DNA-free total RNA. Total RNA (500 ng of total RNA extracted was reverse-transcribed using the High-Capacity cDNA Reverse Transcription Kit (Applied Biosystems) as described by the manufacturer in a C1000 Touch Thermal Cycler (Bio-Rad Laboratories Inc., Hercules, CA, USA). The DyNAmo Flash SYBR Green qPCR kit (Thermo Fisher Scientific, Waltham, MA, USA) was used for real-time PCR using an ABI 7500 Real-Time PCR system (Applied Biosystems, Thermo Fisher Scientific, Waltham, MA, USA). The differences in the cycle number past the threshold (DCt) reflect the differences in the initial template concentration in the tested samples. The ΔΔCt method was used to normalize different transcripts (Table 1) and to calculate fold induction relative to the control. The data were analyzed, and samples were quantified using the Sequence Detection Systems software, version 1.2.3, according to the Relative Quantification Study method (Applied Biosystems). Data were normalized to the *ACTB* housekeeping gene. PCR was carried out at least three times for each sample, and the experiments were performed in triplicate.

### 4.4. Immunofluorescence of Isolated Cells

Freshly isolated KSC, ETA, and LTA cells were fixed with 4% neutral buffered formalin, spotted onto glass slides, and permeabilized with Triton-X100 0.1%. Cells were then incubated with primary antibodies: KRT10 (1:500 dilution) (BioLegend, San Diego, CA, USA), KRT15 (1:200 dilution) (Thermo Scientific, Waltham, MA, USA), involucrin (1:100 dilution) (Sigma-Aldrich, St. Louis, MO, USA), KRT16 (1:100, Serotec, Hercules, CA, USA), STAT3-P (1:100, Cell Signaling Technologym, Danvers, MA, USA), S100A7 (1:100, Santa Cruz, Dallas, TX, USA), and β-defensin (1:100, Abcam, Cambridge, UK). Cells were labeled with anti-mouse or anti-rabbit 488 or 546 Alexa Flour secondary antibodies (Invitrogen, Carlsbad, CA, USA). Nuclei were stained with daimidino-2-phenylindole (DAPI; 1:500, Sigma-Aldich, Saint Louis, MO, USA). Samples were analyzed and images were recorded using a confocal scanning laser microscope (Leica TCS4D; Leica, Exton, PA, USA). Quantification of immunofluorescence staining was performed by analyzing six representative fields for each sample by counting the number of positive cells using the ImageJ software (ImageJ2 version 2.14.0/1.54f) nCounter plugin (Wayne Rasband, National Institutes of Mental Health, Bethesda, MD, USA) or signal intensity. Scores were calculated by counting ± SD.

### 4.5. MTT Assay

Freshly isolated keratinocyte subpopulations were seeded in a 96-well tissue culture plate and treated with IL-17 variants as described, and MTT (3-(4,5-dimethylthiazol-2-yl)-2,5-diphenyltetrazolium bromide, Sigma, St. Loius, MI, USA) assay was performed from 24 to 72 h after treatment, as previously reported [51].

### 4.6. Skin Equivalent Immuno-Histochemistry

Skin equivalents were fixed in 4% buffered PFA. Hematoxylin and eosin (H&E) staining was performed as previously described [51]. Immunohistochemistry was performed using the UltraVision LP Detection System AP Polymer and Fast Red Chromogen assay (Thermo Fisher Scientific, Waltham, MA, USA). The slides were incubated with polyclonal mouse anti-keratin 16 (1:100; Serotec, Hercules, CA, USA) as the primary antibody.

The expression intensity was quantitatively determined using the ImageJ software (ImageJ2 version 2.14.0/1.54f). This was performed in at least 3 different sections, and the results were reported as mean ± standard deviation. Images are representative of three independent experiments performed.

### 4.7. Statistical Analysis

Multiparametric *t*-test or two-way ANOVA was performed by GraphPad Prism 9 (GraphPad Software, La Jolla, CA, USA, www.graphpad.com was used). Significant *p*-values are indicated with *: 0.01 < *p* < 0.05; **: 0.01 < *p* < 0.001; ***0.001 < *p* < 0.0001; **** *p* < 0.0001.

## Figures and Tables

**Figure 1 ijms-26-02989-f001:**
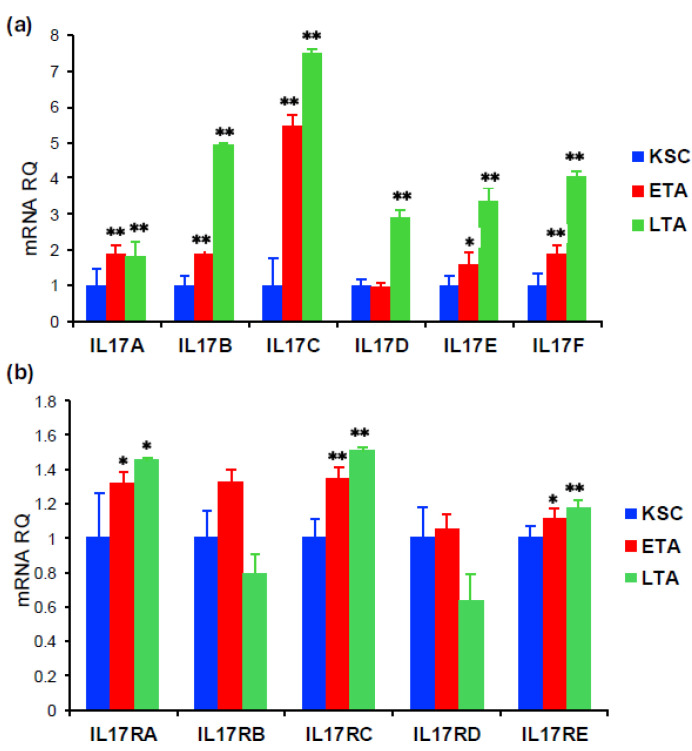
Evaluation of the biological response to IL-17A, IL-17F, and IL-17A/F treatment in keratinocyte subpopulations. (**a**,**b**) mRNA expression of IL-17 variants and receptors evaluated in KSC, ETA, and LTA by qPCR. ACTB was used as housekeeping gene. Statistical analysis was performed using two-way ANOVA. The results are represented as mean + SEM. *p*-values are indicated as the following *: 0.01< *p* <0.05; **: 0.001< *p* <0.01.

**Figure 2 ijms-26-02989-f002:**
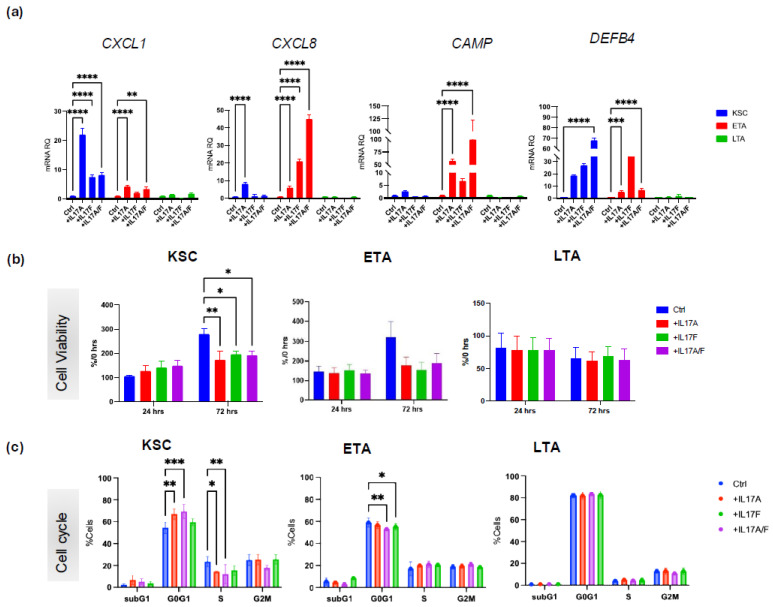
Keratinocyte subpopulations respond to IL-17A, IL-17A/F, and IL-17F. (**a**) mRNA expression of CXCL1, CXCL8, CAMP, and DEFB4 in KSC, ETA, and LTA by qPCR. ACTB was used as the housekeeping gene. (**b**) Cell viability was evaluated in KSC, ETA, and LTA cells after treatment with IL-17A, IL-17A/F, or IL-17F (100ng/mL) the MTT assay, as described in the Materials and Methods section. (**c**) Cell cycle analysis of KSC, ETA, and LTA cells after 72 h of treatment with IL-17A, IL-17A/F, or IL-17F (100 ng/mL) a cytofluorimeter after staining with Nicoletti Solution, as described in the Materials and Methods section. For all experiments, statistical analysis was performed using two-way ANOVA. The results are represented as the mean ± SEM. *p*-values are indicated as the following *: 0.01 < *p* < 0.05; ** 0.001< *p* < 0.01; *** 0.0001< *p* < 0.001; **** *p* < 0.0001.

**Figure 3 ijms-26-02989-f003:**
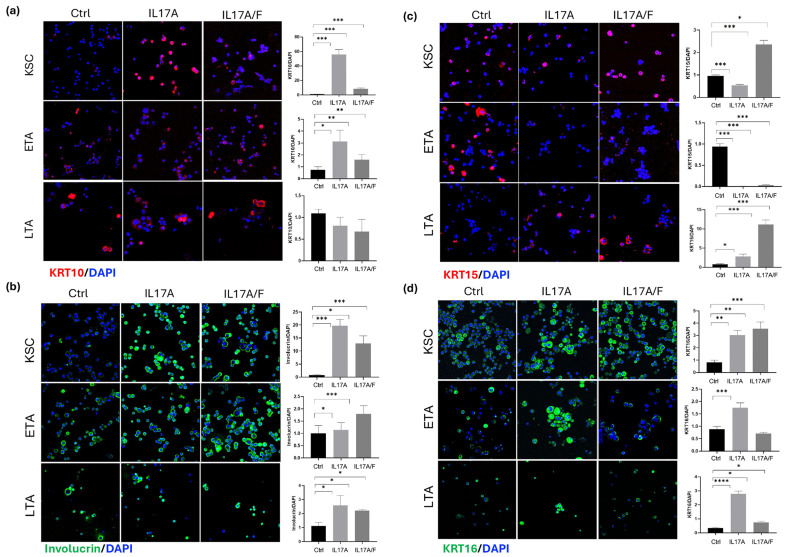
IL-17 variants modulate KRT10, Involucrin, KRT15, and KRT16 expression in keratinocyte subpopulations. (**a**–**d**) Immunofluorescence staining of KRT10, Involucrin, KRT15, and KRT16 in KSC, ETA, and LTA cells after 72-h treatment with IL-17A, IL-17A/F, IL-17F (100 ng/mL). Expression analysis was performed using ImageJ software (ImageJ2 version 2.14.0/1.54f) by normalizing the cell numbers as revealed by DAPI staining and by comparing each treatment with the control. For all results, data are expressed as mean ± SD. Statistical analysis was performed using Student’s *t*-test. *p*-values are indicated as the following *: 0.01 < *p* < 0.05; ** 0.001< *p* < 0.01; *** 0.0001< *p* < 0.001; **** *p* < 0.0001.

**Figure 4 ijms-26-02989-f004:**
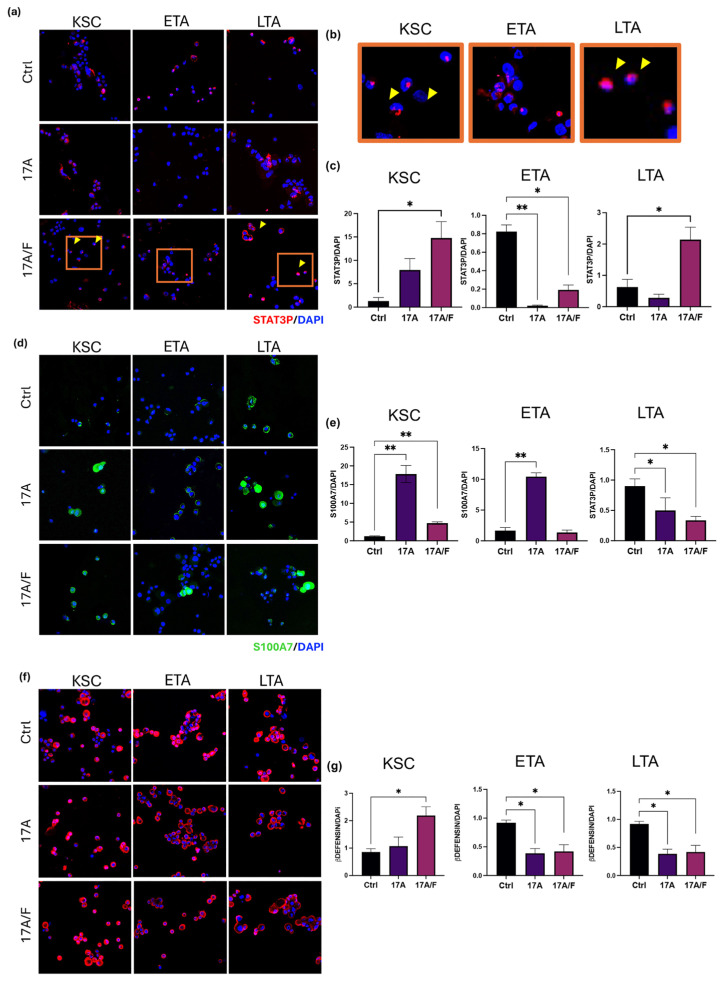
IL-17 variants modulate phospho-STAT3, S100A7 and β-defensin expression in keratinocyte subpopulations. (**a**–**g**) Immunofluorescence staining of phospho-STAT3, S100A7, and β-defensin in KSC, ETA, and LTA cells after 72-h treatment with IL-17A or IL-17A/F (100 ng/mL) or PBS- (Ctrl). Expression analysis was performed using ImageJ software (ImageJ2 version 2.14.0/1.54f) by normalizing the cell numbers as revealed by DAPI staining and by comparing each treatment to the control. For all results, data are expressed as mean ± SD. Statistical analysis was performed using Student’s *t*-test. *p*-values are indicated as the following *: 0.01 < *p* < 0.05; **: 0.01 < *p* < 0.001.

**Figure 5 ijms-26-02989-f005:**
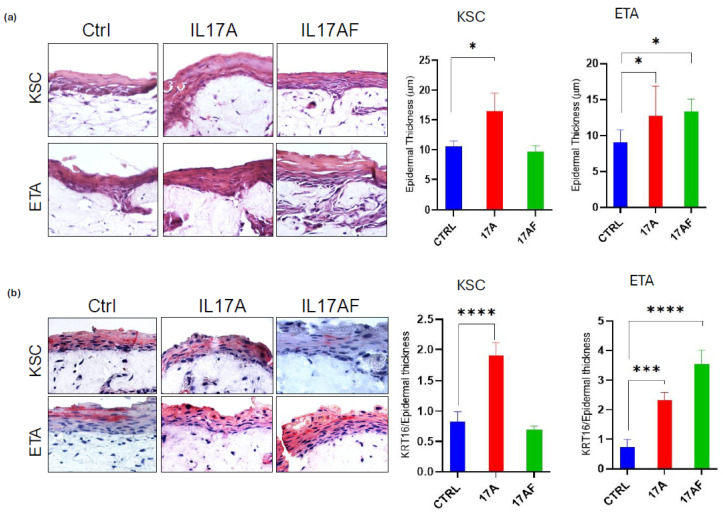
IL-17 variants modulate skin phenotype in 3D reconstruction by increasing epidermal thickness and KRT16 expression (**a**) H&E staining of KSC, ETA, and LTA cell-derived skin reconstructed treated with IL-17A or IL-17A/F (100ng/mL) or PBS- (Ctrl). Measurement analysis was performed using the ImageJ software (ImageJ2 version 2.14.0/1.54f). (**b**) KRT16 immunohistochemical staining of KSC, ETA, and LTA cell-derived skin reconstructed treated with IL-17A or IL-17A/F (100 ng/mL) or PBS- (Ctrl). Signal analysis was performed using the ImageJ software (ImageJ2 version 2.14.0/1.54f). For all results, data are expressed as mean ± SD. Statistical analysis was performed using Student’s *t*-test. *p*-values are indicated as the following *: 0.01 < *p* < 0.05; *** 0.001 < *p* < 0.0001; **** *p* < 0.0001.

**Figure 6 ijms-26-02989-f006:**
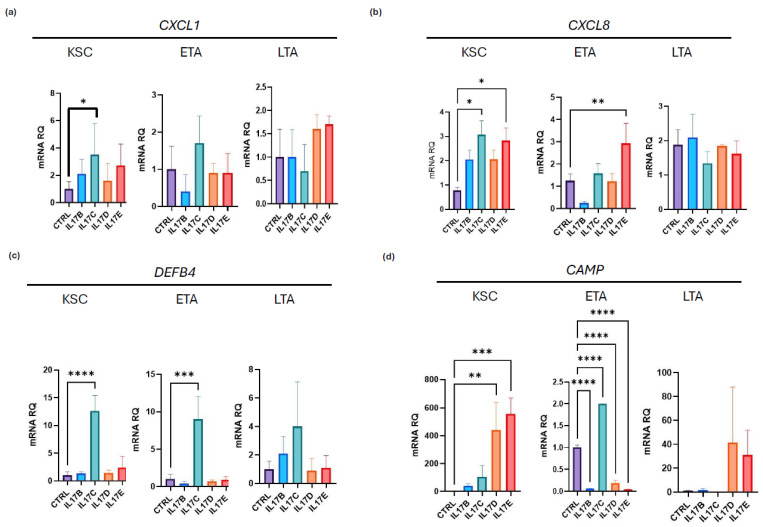
IL-17B, C, D, and E variants modulate CXCL1, CSCL8, DEFB4, and CAMP expression in keratinocyte subpopulations. (**a**–**d**) mRNA expression of CXCL1, CSCL8, DEFB4, and CAMP in KSC, ETA, and LTA by qPCR. ACTB was used as a housekeeping gene. Statistical analysis was performed using two-way ANOVA. The results are represented as mean ± SD. *p*-values are indicated as the following *: 0.01 < *p* < 0.05; **: 0.01 < *p* < 0.001; *** 0.001 < *p* < 0.0001; **** *p* < 0.0001.

**Table 1 ijms-26-02989-t001:** 5 Immunofluorescence of isolated cells.

Gene Symbol		Sequences (5′ > 3′)	Primer (bp)	Amplicon (bp)
*IL-17A*	FP	CTCATTGGTGTCACTGCTACTG	22	78
	RP	CCTGGATTTCGTGGGATTGTG	21	
*IL-17B*	FP	GAGCCCCAAAAGCAAGAGGAA	21	107
	RP	TGCGGGCATACGGTTTCATC	20	
*IL-17C*	FP	CCCTGGAGATACCGTGTGGA	20	236
	RP	GGGACGTGGATGAACTCGG	19	
*IL-17D*	FP	GGGCCAATTTGTGGTTAAGA	20	170
	RP	GCCTCCAGATTGATCTCTGC	20	
*IL-17E* (*IL25*)	FP	CAGGTGGTTGCATTCTTGGC	20	249
	RP	GAGCCGGTTCAAGTCTCTGT	20	
*IL-17F*	FP	CTGGAATTACACTGTCACTTGG	22	108
	RP	GAGATGTCTTCCTTTCCTTGAG	22	
*CXCL1*	FP	AGGGAATTCACCCCAAGAAC	20	130
	RP	ACTATGGGGGATGCAGGATT	20	
*CXCL8* (*IL8*)	FP	GAATGGGTTTGCTAGAATGTGATA	24	129
	RP	CAGACTAGGGTTGCCAGATTTAAC	24	
*DEFB4*	FP	TTGATGTCCTCCCCAGACTC	20	215
	RP	GAGACCACAGGTGCCAATTT	20	
*CAMP*	FP	GCAGTCACCAGAGGATTGTGAC	22	51
	RP	TCAGGCAGCAAATCAAAGGAG	21	
*ACTB*	FP	AAACTGGAACGGTGAAGGTG	20	171
	RP	AGAGAAGTGGGGTGGCTTTT	20	

## Data Availability

Data supporting the findings of this study are available from the corresponding author on reasonable request.

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
