# Peer review of "IL-17 Ligand and Receptor Family Members Are Differentially Expressed by Keratinocyte Subpopulations and Modulate Their Differentiation and Inflammatory Phenotype"

_ijms, 2025, doi:10.3390/ijms26072989_

Round 1
Reviewer 1 Report
Comments and Suggestions for Authors
The main aim of the original article was characteristics of the keratinocyte subpopulations that may play a role in the pathogenesis of psoriasis due to the interaction with Il-17. Thus, in the thicket of theories explaining the development of psoriasis, we return to the well-known regulatory axis, but in a new context. In my opinion, the article is thoughtful and well-organized.
The concise introduction clearly indicates the problem of the work.
It smoothly introduces the topic and explains the validity of the analyses of selected cells and chemokines.
The fact that the results are logically arranged and described is worth noting. Here, each part of the experiment results directly from the previous one. It is good that the authors remind us of the result obtained in the previous one at the beginning of each part. This means that the reader does not get lost in such a large number of results.
Moreover, all figures presented in the text are legible, placed in the right places, complementing the text. Thank you for including the original images as additional materials.
All abbreviations used have been explained. One small note, please enter one shortcut: IL-17AF or IL-17A/F.
Authors to their articles chose 51 references, including 5 from the last fifth years.
In my opinion, the work is much needed, it provides new information about subpopulations of keratinocytes and confirms the great role of IL-17 in the pathogenesis of psoriasis.
Author Response
Comments 1: The main aim of the original article was characteristics of the keratinocyte subpopulations that may play a role in the pathogenesis of psoriasis due to the interaction with Il-17. Thus, in the thicket of theories explaining the development of psoriasis, we return to the well-known regulatory axis, but in a new context. In my opinion, the article is thoughtful and well-organized.
The concise introduction clearly indicates the problem of the work.
It smoothly introduces the topic and explains the validity of the analyses of selected cells and chemokines.
The fact that the results are logically arranged and described is worth noting. Here, each part of the experiment results directly from the previous one. It is good that the authors remind us of the result obtained in the previous one at the beginning of each part. This means that the reader does not get lost in such a large number of results.
Moreover, all figures presented in the text are legible, placed in the right places, complementing the text. Thank you for including the original images as additional materials.
Response: Thank you very much for your thoughtful and detailed feedback on our article. We are truly grateful for your positive comments regarding the clarity and organization of our work, as well as your recognition of the logical flow of the results and the relevance of the figures.
Comments 2: All abbreviations used have been explained. One small note, please enter one shortcut: IL-17AF or IL-17A/F.
Response 2: We appreciate your note, and we modified using a unique shortcut.
Comments 3: Authors to their articles chose 51 references, including 5 from the last fifth years.
In my opinion, the work is much needed, it provides new information about subpopulations of keratinocytes and confirms the great role of IL-17 in the pathogenesis of psoriasis.
Response 3: Thank you very much for your feedback and comments.
Reviewer 2 Report
Comments and Suggestions for Authors
In this manuscript, the authors showed expression profiles and effects of IL-17 ligands and receptor family members in keratinocyte subpopulations. Although this finding provides with interesting information about IL-17 family cytokines, the reviewer has some concerns to be addressed before publishing in IJMS.
Major Comments
1, In main text, but not title, the authors use an inappropriate word “isoforms” as IL-17 ligand (and receptor) family members. The authors have to reconsider the use of “isoforms” in the main text.
2, In main text, the resolution of Fig 2,3,4 was too low and the reviewer could not read any words in graphs.
3, Although the authors compared among the functions of recombinant IL-17A, IL-17F and IL-17A/F heterodimeric proteins in the first experiments of this manuscript, they performed some experiments using only IL-17A and IL-17A/F heterodimeric proteins in latter half (after Result 2.4). The author should justify this point and describe the reason in the result section.
4, In Table 1, primer information about ACTB used as a housekeeping gene.
Minor Comments
1, Abbreviations should be written in parentheses after the full term at their first appearance.
l.98: LTA → late Transit Amplifying (LTA)
2, Gene symbol should be written in italic without “-”, in main text and Table 1
3, In Figure 4 legend l.266 (a-d) → (a-g)?
4, The authors should completely unify the wording. e.g.; The author use both “IL-17A/F” and “IL-17AF” in main text.
5, Typo
l.13 early and differentiated Transit Amplifying → early and late Transit Amplifying
l.454 T-test → t-test
6, In Figure legend mean +- SD → mean ± SD
Author Response
In this manuscript, the authors showed expression profiles and effects of IL-17 ligands and receptor family members in keratinocyte subpopulations. Although this finding provides with interesting information about IL-17 family cytokines, the reviewer has some concerns to be addressed before publishing in IJMS.
Major Comments
Comments 1: 1, In main text, but not title, the authors use an inappropriate word “isoforms” as IL-17 ligand (and receptor) family members. The authors have to reconsider the use of “isoforms” in the main text.
Response 1: We are thankful to the reviewer for the note, we modified the word “isoforms” in the main text, using “variants” instead.
Comments 2: 2, In main text, the resolution of Fig 2,3,4 was too low and the reviewer could not read any words in the graphs.
Response 2: We edited the indicated figures by increasing font size and resolution, as well as the size of each entire figure to fit it on one page together with its legend.
Comments 3: 3, Although the authors compared among the functions of recombinant IL-17A, IL-17F and IL-17A/F heterodimeric proteins in the first experiments of this manuscript, they performed some experiments using only IL-17A and IL-17A/F heterodimeric proteins in latter half (after Result 2.4). The author should justify this point and describe the reason in the result section.
Response 3: Thanks for your comment. We reported “Given the major effect of IL-17A and IL-17A/F on keratinocyte subpopulation cell cycle and viability, we analyzed specific markers associated with epidermal differentiation and proliferation to further understand the meaning of the IL-17A and IL-17A/F-dependent effects on cell cycle and viability.” page 5, line 222 (highlighted in yellow).
Comments 4: 4, In Table 1, primer information about ACTB used as a housekeeping gene.
Response 4: We apologize for the missing information. ACTB has been added to the table.
Comments 5: Minor Comments
1, Abbreviations should be written in parentheses after the full term at their first appearance.
l.98: LTA → late Transit Amplifying (LTA)
2, Gene symbol should be written in italic without “-”, in main text and Table 1
3, In Figure 4 legend l.266 (a-d) → (a-g)?
4, The authors should completely unify the wording. e.g.; The author use both “IL-17A/F” and “IL-17AF” in main text.
5, Typo
l.13 early and differentiated Transit Amplifying → early and late Transit Amplifying
l.454 T-test → t-test
6, In Figure legend mean +- SD → mean ± SD
Response 5: We are thankful to the reviewer for the careful revision, we made all the corrections accordingly.